# FROM EDITOR TO DENSE GEOMETRY ESTIMATOR

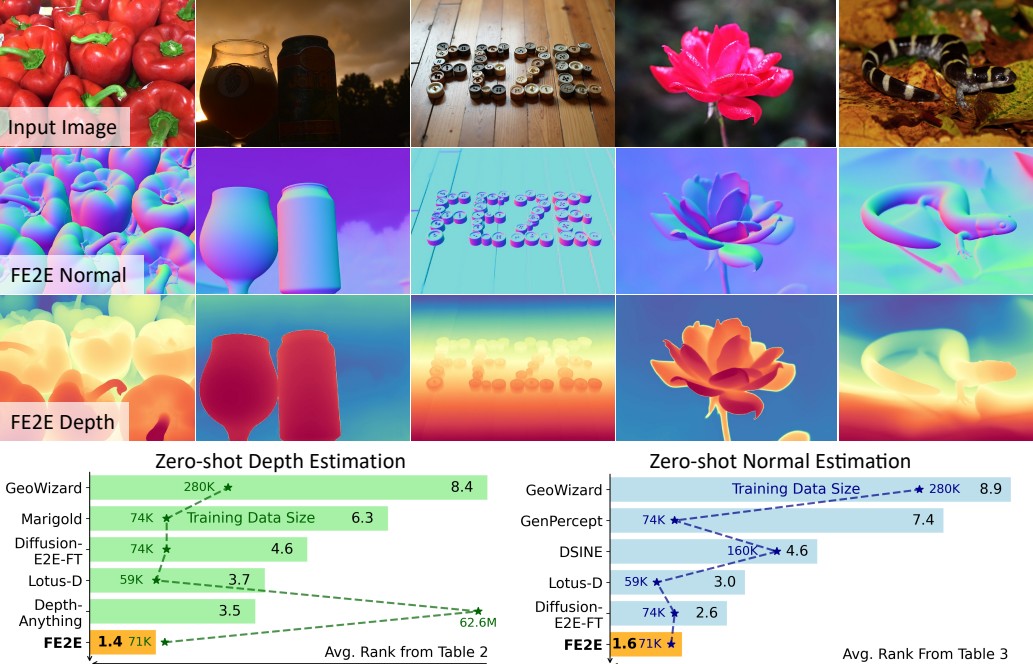

Figure 1: We present **FE2E**, a DiT-based foundation model for monocular dense geometry prediction. Trained with limited supervision, FE2E achieves promising performance improvements in zero-shot depth and normal estimation. Bar length indicates the average ranking across all metrics from multiple datasets, where lower values are better. ★ represents the amount of training data used.

## ABSTRACT

Leveraging visual priors from pre-trained text-to-image (T2I) generative models has shown success in dense prediction. However, dense prediction is inherently an image-to-image task, suggesting that image editing models, rather than T2I generative models, may be a more suitable foundation for fine-tuning. Motivated by this, we conduct a systematic analysis of the fine-tuning behaviors of both editors and generators for dense geometry estimation. Our findings show that editing models possess inherent structural priors, which enable them to converge more stably by "refining" their innate features, and ultimately achieve higher performance than their generative counterparts. Based on these findings, we introduce **FE2E**, a framework that pioneeringly adapts an advanced editing model based on Diffusion Transformer (DiT) architecture for dense geometry prediction. Specifically, to tailor the editor for this deterministic task, we reformulate the editor's original flow matching loss into the "consistent velocity" training objective. And we use logarithmic quantization to resolve the precision conflict between the editor's native BFloat16 format and the high precision demand of our tasks. Additionally, we leverage the DiT's global attention for a cost-free joint estimation of depth and normals in a single forward pass, enabling their supervisory signals to mutually enhance each other. Without scaling up the training data, FE2E achieves impressive performance improvements in zero-shot monocular depth and normal estimation across multiple datasets. Notably, it achieves over 35% performance gains on the ETH3D dataset and outperforms the DepthAnything series, which is trained on 100× data.

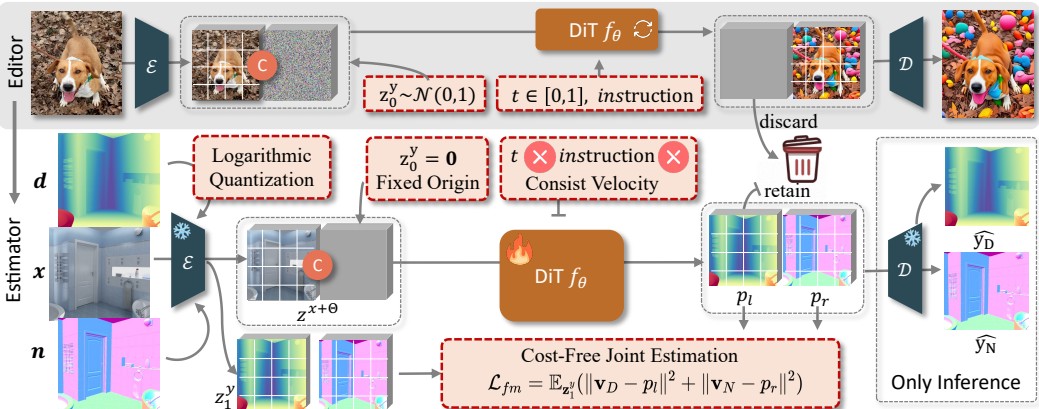

Figure 2: **FE2E Adaptation Pipeline.** The grey background shows the original editor's workflow, while the other details FE2E: ① A pre-trained VAE encodes the logarithmically quantized depth $\mathbf{d}$, input image $\mathbf{x}$, and normals $\mathbf{n}$ into latent space. ② The DiT $f_\theta$ learns a constant velocity $\mathbf{v}$ from a fixed origin $\mathbf{z}_0^y$ to the target latent $\mathbf{z}_1^y$, independent of $t$ or instructions. ③ By repurposing the discarded output region, FE2E jointly predicts depth and normals without extra computation. Training loss is computed in the latent space, with final predictions decoded by VAE only at inference.

# 1 INTRODUCTION

Dense geometry prediction tasks, such as depth/normal estimation, are crucial for a wide range of applications such as augmented reality (Minaee et al., 2022), and 3D reconstruction (Li et al., 2025). Estimating pixel-level geometric attributes from a single image is an ill-posed problem and can only be solved with the help of prior knowledge, such as typical object shapes and sizes, occlusion patterns, etc. Based on this observation, recent works ingeniously leverage the priors from pre-trained text-to-image (T2I) generators, typically Stable Diffusion (Rombach et al., 2022), for zero-shot dense prediction (Ke et al., 2024), yielding impressive results with limited training data.

However, these generative models are initially designed for T2I generation and lack the ability to capture the geometric cues from the absent image inputs. In contrast, image editing models have recently risen to be a universal framework to solve more diversified image-to-image (I2I) tasks, such as semantic segmentation and depth estimation (Wu et al., 2025). We argue that these editing models not only align with the dense estimation paradigm but also possess a deep understanding of input images while maintaining the generative advantages, and offer a more suitable foundation for dense geometry prediction.

Motivated by this intuition, we systematically analyze the fine-tuning process of the image editing models versus their generative counterparts. Our analysis reveals that the features of editing models are inherently aligned with geometric structures, and the fine-tuning process only requires to "refine" and "focus" this perceptual ability for dense estimation tasks. In contrast, although generative models can gradually acquire this capability from scratch, this process leads to substantial feature reshaping and cannot fundamentally bridge this gap (Sec. 3.1). Therefore, in this paper, we explore this editing option and propose **From Editor to Estimator (FE2E)** (Fig. 2), a diffusion transformer (DiT) model built upon the current SoTA editor Step1X-Edit (Liu et al., 2025), along with a fine-tuning protocol to adapt it for dense geometry prediction tasks.

However, the direct adaptation of an image editing model for geometric dense prediction is often suboptimal, due to the inherent differences between the two tasks. First, compared to editing, the dense geometry prediction is more deterministic, as only one unique ground-truth (GT) exists. Our analysis of Step1X-Edit reveals that its training objectives—predicting *instantaneous velocity*[1]—are only tailored for indeterminate tasks and will introduce errors in dense prediction. Based on this observation, we reformulate the training objective as a *consistent velocity*[1] and set a fixed starting point for stable training. Second, image editing models like Step1X-Edit are typically trained with BF16 precision, which is sufficient for RGB outputs, whereas dense geometry prediction tasks like

---

[1]velocity denotes the direction and speed of transformation that "change input to output", see Sec. 3.2

depth estimation demand much higher numerical precision. This discrepancy does not occur in previously adopted foundation models like Stable Diffusion v1.5/v2, which offer FP32 checkpoints. To address this limitation and improve computational efficiency, we analyze the GT quantization strategies and adopt a logarithmic quantization to alleviate precision-related artifacts.

After adapting the editor, we now focus on its role as an estimator. In this paper, we select two primary geometry prediction tasks: zero-shot depth estimation and normal estimation, to validate FE2E's effectiveness. Depth and normal both contribute to a unified geometric representation, and joint estimation can leverage their potential connections. Unlike GeoWizard(Fu et al., 2024) that introduces additional cross-attention and switchers, we observe that the global attention mechanism of DiT can be repurposed to perform joint estimation of depth and surface normals in a single forward pass, incurring virtually no additional computational cost. This design allows the supervisory signals from both tasks to interact, enhancing the overall performance.

Extensive experiments demonstrate that our model achieves notable zero-shot performance improvements compared to previous state-of-the-art (SoTA) models. Our contribution can be summarized as follows:

- We systematically analyze the fine-tuning process of image editors and generators, revealing that editing models are more suitable for dense geometry prediction. Based on this, we introduce FE2E, a novel framework that, for the first time, successfully adapts a pre-trained image editing model for this task.
- We identify and address the challenges that arise from this paradigm shift: 1) Reformulate the training objective to align with the deterministic nature of dense prediction; 2) Adopt a logarithmic quantization to resolve the precision conflict. 3) Design cost-free joint estimation to allow supervision from different tasks to mutually enhance predictions.
- Based on above enhancements, FE2E achieves impressive performance gains, including **35%** AbsRel improvement on the ETH3D dataset. Even when using only **0.2%** of the geometric GT data for training, FE2E outperforms the data-driven models like DepthAnything v1/v2.

## 2 RELATED WORK

**Image Generative and Editing models** In the field of image generation, Stable Diffusion series (Rombach et al., 2022) and FLUX series (Labs, 2024) models have basically become the community standard. They both trained with massive datasets and demonstrate extremely high generation quality. Meanwhile, the field of image editing is also evolving rapidly. Recent advancements include Step1X-Edit (Liu et al., 2025), a model fine-tuned from FLUX demonstrating superior instruction-following and image understanding capability; The multi-modal Qwen-Image (Wu et al., 2025) Editor combines with LLM, attempting to expand the editor into a unified computer vision framework; The concurrent work FLUX-Kontext (Labs et al., 2025) unifies editing and generation with robust character consistency. We conduct a more detailed review in the appendix Sec E.

**Dense Geometry Estimation,** encompassing tasks like depth and normal estimation(Wang et al., 2024a;b), is a cornerstone of 3D computer vision. Early research predominantly focused on supervised learning paradigms, where models were trained and evaluated on specific datasets (Eigen et al., 2014; Eigen & Fergus, 2015). A significant shift occurred with MiDaS (Ranftl et al., 2020), which pioneered cross-dataset generalization for dense estimation. This line of work was extended by models like DPT (Ranftl et al., 2021) and Omnidata (Eftekhar et al., 2021), which further improved zero-shot performance. More recently, the field has witnessed the rise of data-driven models such as the Depth Anything series (Yang et al., 2024a;b) and the Metric3D series (Hu et al., 2024), which leverage massive datasets to train powerful, general-purpose geometric estimators.

**Generative Models for Dense Estimation.** In parallel to the trend of scaling up data, an alternative approach emerged by leveraging the rich priors of pre-trained generative models. Works like Marigold (Ke et al., 2024) and GeoWizard (Fu et al., 2024) showed that fine-tuning diffusion models on limited data could yield remarkable performance, effectively harnessing the models' learned world knowledge. This paradigm was further refined by GenPercept (Xu et al., 2024), StableNormal (Ye et al., 2024), Diffusion-E2E-FT (Martin Garcia et al., 2025), Lotus (He et al., 2024), and Jasmine (Wang et al., 2025). These studies identified and addressed the limitations of standard diffusion formulations, developing the single-step denoising architecture to boost performance. In this paper, we also build FE2E with limited data, posit that the I2I editing models are inherently better than the T2I models (Stable Diffusion (Rombach et al., 2022)) for dense estimation.

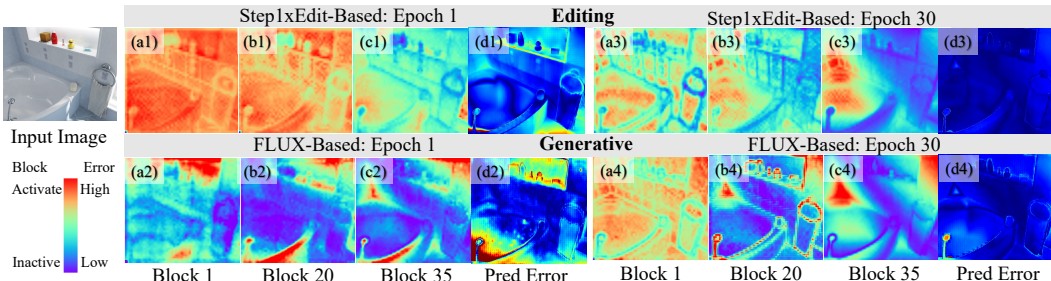

Figure 3: **Comparison between the Generative and Editing foundation models.** We analyze the feature evolution at both the initial (Epoch 1) and final (Epoch 30) stages of fine-tuning, resulting in 4 groups. Each group presents: the DiT features at the input end (Block1), middle layers (Block20), output end (Block35), and the depth prediction's AbsRel (Absolute Relative error). Visual implementation detailed in Sec B.

## 3 METHODS

We first conduct an investigation into the fine-tuning process between the editor and generator in Sec 3.1. Then, we adapt the editor into an estimator (Sec 3.2) by introducing three key contributions to the training objective (Sec 3.3), GT quantization(Sec 3.4), and joint estimation (Sec 3.5).

### 3.1 FINE-TUNING ANALYSIS OF EDITOR AND GENERATOR

In this paper, we select Step1X-Edit as the editor and FLUX as the generator, owing to their shared DiT architecture and SoTA performance in their respective tasks. Note that FE2E can also generalize to other DiT-based editing models (see Sec 4.4). To facilitate a fair comparison, we adopt the *improved* experimental setup and additionally train a FLUX-based estimator for this analysis. The training details are provided in Sec B. Through this systematic analysis, we identify **three** key advantages of editing-based models over generative predecessors for dense estimation tasks.

First, the editing model possesses a superior inductive bias for image-to-image dense estimation tasks, providing a much stronger starting point for finetuning. This is evident in Fig. 3 (a1 *v.s.* a2), in the early stage and blocks, the

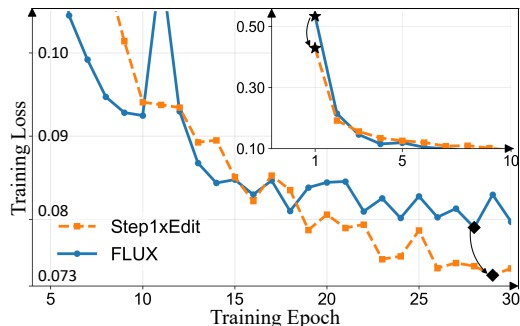

Figure 4: **Quantitative comparison of the training loss** between Generative and Editing foundation models. The main plot details the *convergence* loss from epoch 5 to 30, while the inset displays the steep *initial* loss reduction during the first 10 epochs, which occurs on a different scale.

editor's internal features already align with the input image's geometric structures, while the generative ones are abstract and unstructured. The loss difference in Fig. 4 ★ shows the same conclusion.

Second, the above difference directly impacts the learning dynamics: as illustrated in Fig. 4, the editor achieves a more stable convergence, in contrast to the oscillations seen in the generative ones. This difference can be further explained in Fig. 3, the fine-tuning process significantly reshapes the characteristics of generative models, while the editor's features are more like a "refinement" and "focusing". After 30 epochs of fine-tuning, the generative model learned highly structured and semantic features (a4, b4, c4) from chaotic states (a2, b2, c2), achieving a qualitative leap. Whereas the editing ones make the well-structured features (a1, b1, c1) clearer and task-oriented (a3, b3, c3), with their features being incrementally honed rather than fundamentally altered.

Third, the "structured learning" and "characteristics reshaping" mentioned above are unable to address the shortcomings of the generative model. As shown in Fig. 4 (epochs 20-30, especially ◆), the generative model's training loss meets a bottleneck around 0.08, while the editing ones can reduce to 0.073. The Table 4 (ID6 vs. ID7) further verifies that this bottleneck persists at test time, resulting in a significant performance gap.

In summary, the analysis of feature evolution, training dynamics, and performance consistently demonstrates that editing models provide a more *stable*, *effective*, and *promising* foundation for dense geometry estimation.

## 3.2 FROM EDITOR TO ESTIMATOR

As shown in Fig. 2, we pose monocular dense estimation as an image editing task and use the **Flow Matching** Loss for supervision.

Initially, we take the input image $\mathbf{x} \in \mathbb{R}^{H \times W \times 3}$ as the editing source and the geometric annotation $\mathbf{y} \in \mathbb{R}^{H \times W \times 3}$ as the expected editing results. First, the VAE, which consists an encoder $\mathcal{E}(\cdot)$ and a decoder $\mathcal{D}(\cdot)$, is used to encode the input image $\mathbf{x}$ into a latent representation $\mathbf{z}^x = \mathcal{E}(\mathbf{x}) \in \mathbb{R}^{h \times w \times c}$. Then, the editing process is modeled as a flow path from a noise vector $\mathbf{z}_0^y \sim \mathcal{N}(0, \mathbf{I})$ to the target latent representation $\mathbf{z}_1^y = \mathcal{E}(\mathbf{y})$. The trajectory is defined as:

$$\mathbf{z}_t^y = t\mathbf{z}_1^y + (1-t)\mathbf{z}_0^y, \quad t \in [0, 1]. \tag{1}$$

The DiT backbone, denoted as $f_\theta$, is trained to predict the **velocity vector** of this flow, which is simply $\mathbf{v} = \frac{d\mathbf{z}_t^y}{dt} = \mathbf{z}_1^y - \mathbf{z}_0^y$. The model is optimized by minimizing the flow matching loss (Lipman et al., 2022):

$$\mathcal{L} = \mathbb{E}_{t, \mathbf{z}_1^y, \mathbf{z}_0^y} \|\mathbf{v} - f_\theta(\mathbf{z}^x, \mathbf{z}_t^y, t)\|^2. \tag{2}$$

In the inference stage, we predict the editing target $\hat{\mathbf{z}}_1^y$ by solving the following ordinary differential equation:

$$\hat{\mathbf{z}}_1^y = \mathbf{z}_0^y + \int_0^1 f_\theta(\mathbf{z}^x, \mathbf{z}_t^y, t)dt, \tag{3}$$

and the final dense geometry predictions are given by $\hat{\mathbf{y}} = \mathcal{D}(\hat{\mathbf{z}}_1^y)$. More details of the flow matching process are provided in Sec D.

## 3.3 CONSISTENT VELOCITY FLOW MATCHING WITH DETERMINISTIC DEPARTURE

Flow Matching has been widely adopted in modern generative and editing modeling. As shown in Fig. 5, previous work Mean-Flow (Geng et al., 2025) has identified that, since the model learns the velocity over all possible flow paths in Eq 1, the global instantaneous velocity field becomes inherently non-linear and typically induces a curved trajectory.

During inference, as shown in Fig. 5 (b), the ideal integration path in Eq. 3 is approximated by a discrete numerical solver, which introduces a non-trivial approximation error for high-precision tasks such as dense geometric estimation.

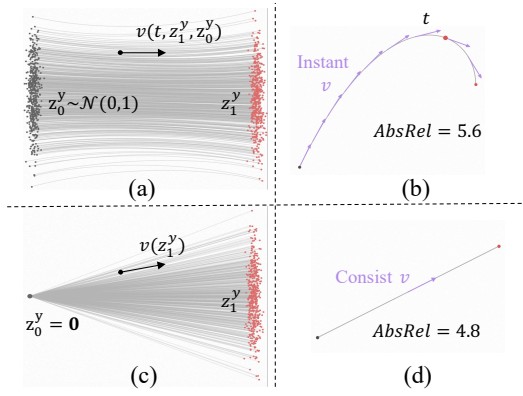

Figure 5: **Left**: GT velocity field for network training. The gray dots represent different Gaussian noise (top) or **zero** starting point (bottom), the red dots represent data samples. **Right**: Instantaneous velocity $v$ determines the tangent direction and creates errors in the cumulative path (top); The constant speed path is a straight line.

An intuitive idea is to find a straight integration path, which means the velocity direction always remains the same. In this paper, we further require the velocity magnitude to be consistent so that the velocity is completely independent of $t$, and redefine the loss as:

$$\mathcal{L} = \mathbb{E}_{\mathbf{z}_1^y, \mathbf{z}_0^y} \|\mathbf{v} - f_\theta(\mathbf{z}^x, \mathbf{z}_0^y)\|^2. \tag{4}$$

Additionally, one key characteristic of generative/editing models is their stochastic nature, which is essential for producing diverse outputs. For deterministic dense prediction tasks, however, this stochasticity is not only unnecessary but also introduces undesirable variance into the training objective, complicating the optimization of $f_\theta$. Therefore, we simplify the objective by reducing it from

a stochastic expectation over all possible $\mathbf{z}_0^y$ to a deterministic formulation with a fixed $\mathbf{z}_0^y = \mathbf{0}$. This further refines the loss as:

$$\mathcal{L} = \mathbb{E}_{\mathbf{z}_1^y} \| \mathbf{v} - f_\theta(\mathbf{z}^x) \|^2, \tag{5}$$

and the inference process can be simplified as:

$$\mathbf{z}_1^y = \mathbf{z}_0^y + \int_0^1 f_\theta(\mathbf{z}^x) dt = \mathbf{0} + (1-0)f_\theta(\mathbf{z}^x) = f_\theta(\mathbf{z}^x). \tag{6}$$

Overall, as shown in Fig. 5 (c) and (d), our refined flow matching not only eliminates the errors introduced by discretized curved trajectories and random starting points, but also significantly reduces inference time, achieving simultaneous improvements in both performance and efficiency.

### 3.4 LOGARITHMIC ANNOTATION QUANTIZATION

Modern generative/editing models are almost exclusively trained with BF16 precision. This is not only because BF16 precision saves the training cost, but also for their typical outputs, RGB images; this precision is entirely sufficient. Specifically, a normalized BF16 value is represented as:

$$V = (-1)^S \times 2^{(E-127)} \times (1.F)_2,$$

where S is the sign (1 bit), E is the exponent (8 bits), and F is the fraction (7 bits). Due to the [-1,1] data range of VAE encoded input (Liu et al., 2025), the worst-case precision occurs at $\pm[0.5, 1.0]$, which is $2^{126-127} \times 2^{-7} = 1/256$ and perfectly satisfies the RGB range of 0-255.

Uncritically finetuning these models with FP32, as done in Marigold or Lotus, not only increases training/inference costs, but also leads to suboptimal inheritance of the baseline model's priors, and restricts the capabilities of BF16-only models like Step1X-Edit. Therefore, finetuning with BF16 is necessary.

Table 1: Quantization errors at BF16 precision on Virtual KITTI dataset. Calculation details in Sec C.

|  | (a) Uniform | | (b) Inverse | | (c) Logarithmic | |
|---|---|---|---|---|---|---|
|  | Error | Absrel | Error | Absrel | Error | Absrel |
| 80m | 16cm | 0.002 | 125m | 1.563 | 1.04m | 0.013 |
| 0.1m | 16cm | 1.600 | 0.2mm | 0.002 | 1.3mm | 0.013 |

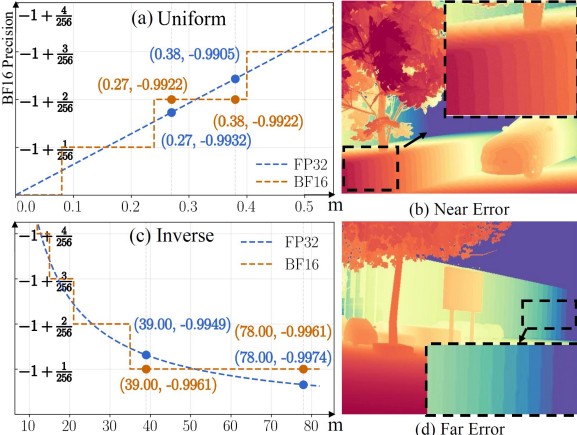

(b) Near Error

(d) Far Error

Figure 6: Illustration of BF16 quantization error. (b) and (d) show GT depth visualized with BF16 precision. For clarity, (a) and (c) use Maigold's VAE regularization, mapping max/min values to 1/-1, respectively.

However, as shown in Fig. 6 (a,b), when meeting the depth annotations in the Virtual KITTI dataset, the valid depth range is 0-80m. **Uniformly** regularizing to [-1,1] requires a reduction of 40 times, and the accuracy of 1/256 is reflected in the original depth with a significant error of $40/256 \approx 0.16$m. These errors result in an AbsRel of 1.6 at 0.1m (Table 1 (a)) and make the finetune process unfeasible. Previous works have employed an **inverse** quantization scheme, which means converting the reciprocal of the depth, or disparity, to BF16 precision (Fig. 6 (c, d)). As shown in Table 1 (b), despite offering extremely high precision at close ranges, this scheme becomes entirely unusable at greater distances, and even makes 39m and 78m correspond to the same value. The principles and calculations are detailed in Sec C.

After numerous attempts and explorations, we use the **logarithmic** depth quantization to achieve good precision at both near and far ranges (Table 1(c)) while reducing training and inference costs. Specifically, we first perform the logarithmic quantization with $D_{log} = ln(D_{GT} + 1e - 6)$, then follow and refine Marigold's depth normalization strategy, defining the supervision label $\mathbf{y}_D$ as:

$$\mathbf{y}_D = \left\langle \left( \frac{(D_{log} - D_{log,2})}{(D_{log,98} - D_{log,2})} - 0.5 \right) \times 2 \right\rangle, \tag{7}$$

where $D_{log,i}$ corresponds to the $i\%$ percentiles of $D_{log}$, and $\langle \cdot \rangle$ is the BF16 precision truncation.

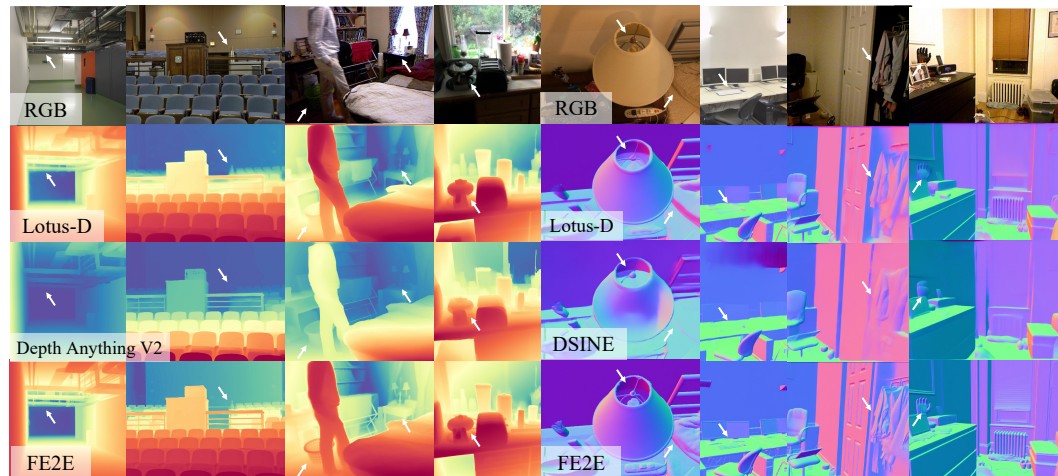

Figure 7: **Quantitative comparison on zero-shot depth and normal estimation.** The 1st row shows the input, the 2nd, 3rd rows are previous SoTA methods results, and the 4th row is ours prediction. White arrows highlight the regions we significantly improved. Zoom in for better view.

### 3.5 COST-FREE JOINT ESTIMATION

There are inherent connections between depth and normal, as both contribute to a unified geometric representation of 3D shapes. Normals describe surface variations and undulations, while depth outlines the spatial arrangement that guides the orientation of the normals. Thus, previous work like GeoWizard (Fu et al., 2024) attempted to control the SD model with a geometry switcher, but this approach doubled the training cost.

In contrast, as shown in Fig. 2 grey part, Step1X-Edit and other DiT-based editing works have found that, the DiT architecture can effectively guide image generation by horizontally concatenating the noise and condition latents, that is, the input is formulated as $z^{x+\Theta} = \text{concat}(z^x, z^\Theta) \in \mathbb{R}^{h \times 2w \times c}$ ($z^\Theta$ is the noise latents, shown in Fig. 2). However, after processing by the DiT, although the model's output has the same shape as the input, $f_\theta(z^{x+\Theta}) = [p_l, p_r] \in \mathbb{R}^{h \times 2w \times c}$, supervision is only applied to the region corresponding to the original noise, *i.e.*, $\mathcal{L} = \|\mathbf{v} - p_r\|^2$, where $p_l, p_r \in \mathbb{R}^{h \times w \times c}$.

It is worth noting that DiT architecture possesses the global attention across the $(h, 2w)$ dimension, which naturally allows for mutual information exchange between $p_l$ and $p_r$. Based on this observation, without introducing any additional training or inference costs, we further incorporate another task's supervision on $p_l$ during finetuning, extending Eq. 5 for both tasks ($\mathbf{v}_D/\mathbf{v}_N$ are velocity training objectives for depth/normal task, respectively) as:

$$\mathcal{L}_{fm} = \mathbb{E}_{\mathbf{z}_1^y}(\|\mathbf{v}_D - p_l\|^2 + \|\mathbf{v}_N - p_r\|^2). \tag{8}$$

## 4 EXPERIMENTS

### 4.1 IMPLEMENTATION DETAILS

We build FE2E upon the Step1X-Edit v1.0 framework (Liu et al., 2025). To further enhance the DiT's representational power, we also introduce an auxiliary dispersion loss that encourages features from different samples to spread out in the hidden space, which is detailed in Sec A.1. During finetuning, all parameters except for the DiT module are frozen, and the language control input is left blank. The process employs LoRA (Hu et al., 2021) with rank $= 64$ and scale factor $\alpha = 32$. We trained for 30 epochs using the AdamW optimizer (Loshchilov & Hutter, 2019) with an initial learning rate of $1 \times 10^{-4}$. With gradient checkpoint enabled, the model can be trained on a single RTX 4090 GPU, but to accelerate experimentation, training was conducted on NVIDIA H20 GPUs, completing in approximately 1.5 days.

### 4.2 TRAINING DATASETS

We train our model for joint depth and normal estimation on a mixture of two synthetic datasets: Hypersim (Roberts et al., 2021) and Virtual KITTI (Cabon et al., 2020). For **Hypersim**, a photorealistic indoor dataset, we use its official training split after filtering out samples with over 1% invalid pixels,

Table 2: **Quantitative comparison on zero-shot affine-invariant depth estimation** between FE2E and SoTA methods. The best and second best performances are highlighted. * denotes the method relies on pre-trained Stable Diffusion.

| Method | Training Data↓ | NYUv2 (Indoor) AbsRel↓ | δ1↑ | KITTI (Outdoor) AbsRel↓ | δ1↑ | ETH3D (Various) AbsRel↓ | δ1↑ | ScanNet (Indoor) AbsRel↓ | δ1↑ | DIODE (Various) AbsRel↓ | δ1↑ | Avg Rank↓ |
|---|---|---|---|---|---|---|---|---|---|---|---|---|
| MiDaS | 2M | 11.1 | 88.5 | 23.6 | 63.0 | 18.4 | 75.2 | 12.1 | 84.6 | 33.2 | 71.5 | 10.6 |
| GeoWizard | 280K | 5.6 | 96.3 | 14.4 | 82.0 | 6.6 | 95.8 | 6.4 | 95.0 | 33.5 | 72.3 | 8.4 |
| GenPercept | 74K | 5.6 | 96.0 | 13.0 | 84.2 | 7.0 | 95.6 | 6.2 | 96.1 | 35.7 | 75.6 | 7.8 |
| Marigold$_{v1.1}$ * | 74K | 5.8 | 96.1 | 11.0 | 88.8 | 7.0 | 95.5 | 6.6 | 95.3 | 30.4 | 77.3 | 7.6 |
| Marigold* | 74K | 5.5 | 96.4 | 9.9 | 91.6 | 6.5 | 95.9 | 6.4 | 95.2 | 30.8 | 77.3 | 6.3 |
| DepthAnything V2 | 62.6M | 4.5 | 97.9 | 7.4 | 94.6 | 13.1 | 86.5 | - | - | 26.5 | 73.4 | 5.4 |
| Lotus-G* | 59K | 5.4 | 96.8 | 8.5 | 92.2 | 5.9 | 97.0 | 5.9 | 95.7 | 22.9 | 72.9 | 4.7 |
| Diffusion-E2E-FT* | 74K | 5.4 | 96.5 | 9.6 | 92.1 | 6.4 | 95.9 | 5.8 | 96.5 | 30.3 | 77.6 | 4.6 |
| Lotus-D* | 59K | 5.1 | 97.2 | 8.1 | 93.1 | 6.1 | 97.0 | 5.5 | 96.5 | 22.8 | 73.8 | 3.7 |
| DepthAnything | 62.6M | 4.3 | 98.1 | 7.6 | 94.7 | 12.7 | 88.2 | 4.3 | 98.1 | 26.0 | 75.9 | 3.5 |
| **FE2E** | **71K** | **4.1** | **97.7** | **6.6** | **96.0** | **3.8** | **98.7** | **4.4** | **97.5** | **22.8** | **81.2** | **1.4** |

Table 3: **Quantitative comparison on zero-shot surface normal estimation** between FE2E and SoTA methods. ‡refers to the Marigold normal model as detailed in their repository.

| Method | Training Data↓ | NYUv2 (Indoor) MeanErr↓ | 11.25°↑ | ScanNet (Indoor) MeanErr↓ | 11.25°↑ | iBims-1 (Indoor) MeanErr↓ | 11.25°↑ | Sintel (Outdoor) MeanErr↓ | 11.25°↑ | Avg. Rank |
|---|---|---|---|---|---|---|---|---|---|---|
| Marigold$^{‡}$* | 74K | 20.9 | 50.5 | 21.3 | 45.6 | 18.5 | 64.7 | - | - | 9.5 |
| GeoWizard* | 280K | 18.9 | 50.7 | 17.4 | 53.8 | 19.3 | 63.0 | 40.3 | 12.3 | 8.9 |
| GenPercept* | 74K | 18.2 | 56.3 | 17.7 | 58.3 | 18.2 | 64.0 | 37.6 | 16.2 | 7.4 |
| StableNormal* | 250K | 18.6 | 53.5 | 17.1 | 57.4 | 18.2 | 65.0 | 36.7 | 14.1 | 7.2 |
| Lotus-G* | 59K | 16.5 | 59.4 | 15.1 | 63.9 | 17.2 | 66.2 | 33.6 | 21.0 | 5.2 |
| DSINE | 160K | 16.4 | 59.6 | 16.2 | 61.0 | 17.1 | 67.4 | 34.9 | 21.5 | 4.6 |
| Lotus-D* | 59K | 16.2 | 59.8 | 14.7 | 64.0 | 17.1 | 66.4 | 32.3 | 22.4 | 3.0 |
| Diffusion-E2E-FT* | 74K | 16.5 | 60.4 | 14.7 | 66.1 | 16.1 | 69.7 | 33.5 | 22.3 | 2.6 |
| Marigold$_{v1.1}$ * | 77K | 16.1 | 60.5 | 14.5 | 66.1 | 16.3 | 68.5 | - | - | 2.0 |
| **FE2E*** | **71K** | **16.2** | **59.6** | **13.8** | **67.2** | **15.1** | **70.6** | **31.2** | **22.3** | **1.6** |

resulting in approximately 51k images at a 1024×768 resolution. For **Virtual KITTI**, a synthetic street view dataset, we utilize four driving scenarios, totaling around 20k samples at a 1216×352 resolution with a maximum depth of 80m. Following Marigold, each training batch is constructed by sampling from Hypersim and Virtual KITTI with probabilities of 90% and 10%, respectively.

The evaluation dataset and evaluation metrics also follow Marigold and are detailed in Sec A.2, A.3, respectively.

## 4.3 QUANTITATIVE EVALUATION

**Zero-shot Depth Estimation Comparison** As presented in Table 2, FE2E significantly outperforms recent SoTA methods across five challenging benchmarks. Notably, on the ETH3D and KITTI datasets, it reduces the AbsRel error by **35%** and **10%** respectively, compared to the 2nd-best method. Remarkably, despite being trained on only **0.071M** images, FE2E's average rank surpasses that of the DepthAnything series, which was trained on a massive **62.6M** image dataset. This highlights the effectiveness of our strategy: inheriting the editing model priors rather than simply scaling up training data. Furthermore, qualitative comparisons in Fig. 1 and 7 demonstrate that FE2E produces superior results in challenging lighting conditions (extreme-light, low-light, etc.) and better preserves distant details, which reveal the core advantages that contribute to FE2E's superior performance. We provide further comparisons with concurrent unified works in Sec F.

**Zeroshot Normal Estimation Comparison** As presented in Table 3, FE2E also achieves SoTA performance on the zero-shot normal estimation task, outperforming the methods in the recent 2 years across four benchmarks. This quantitative superiority stems from its ability to handle complex geometries. As illustrated in Fig. 1, 7, FE2E excels at reconstructing intricate details such as surface folds and small objects, which are often challenging for other models.

## 4.4 ABLATION STUDY

**Effect of Foundation Model.** FE2E is based on the new foundation model Step1X-Edit, the direct adaptation protocol (ID2) establishes a strong baseline, outperforming Marigold (based on SD v2)

by 8% and 4% on ETH3D and KITTI datasets, respectively (AbsRel, the same below). Building on this, FE2E (ID8) further reduces the AbsRel by 32.1% on ETH3D and 30.5% on KITTI, which confirms the effectiveness of our proposed techniques.

**Effect of Editing Priors.** We trained the FLUX-based model under both *DirectAdapt* and *Improved* settings, corresponding to ID1 and ID7 in Table 4. Compared with their counterparts (ID2 and ID6), the editing-based models consistently outperform the generative models, regardless of equipping our proposed improvements. These results, together with the findings in Sec 3.1, highlight the effectiveness of leveraging editing model priors for dense prediction tasks.

**Effect of Improved Flow Matching.** Adopting the consistent velocity training objective effectively eliminates accumulated inference errors from the original paradigm, leading to notable performance gains of 7% on KITTI and 10% on ETH3D (ID2 *v.s.* ID3). Introducing a fixed starting point further eases optimization and brings additional improvements (ID3 *v.s.* ID4).

**Effect of Data Quantization.** Supervision leads to substantial performance gains, with ID6 outperforming ID4 by 19% and 13% on the KITTI and ETH3D datasets, respectively. Notably, inverse quantization (ID5) generally outperforms uniform quantization, typically because there are more valid pixels nearby.

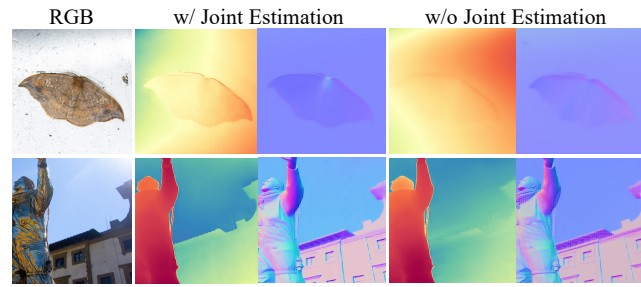

RGB — w/ Joint Estimation — w/o Joint Estimation

Figure 8: Qualitative comparison on the Joint Estimation. The 'w/o Joint Estimation' shows 2 models' results.

**Effect of Joint Estimation.** Results from ID6 and ID8 demonstrate that joint prediction further enhances model performance. This synergistic effect is more clearly illustrated in Fig. 8, where joint training yields notable improvements in challenging scenarios such as flat butterfly structures and distant buildings.

**Extensibility to Other Editors.** We apply our adaptation protocol to the concurrent FLUX-Kontext model. The finetuned model (ID9) achieves comparable or even superior performance to its Step1X-Edit counterpart (ID8), likely due to the stronger editing priors in FLUX-Kontext, which confirms the broad applicability and high potential of our approach.

Table 4: **Ablation studies** of our adaptation protocol. Here we show the results in depth estimation. CV: Consistent Velocity; FS: Fixed Start; JE: Joint Estimation; Quant: Quantization, sub-items same with Table 1). *DirectAdapt* refers to the formulation in Sec 3.2, and *Improved* setting denotes all improvements except JE (output dimensions of FLUX cannot support JE).

| ID | Note | Foundation | CV | FS | JE | Quant | KITTI AbsRel↓ | KITTI δ1 ↑ | ETH3D AbsRel↓ | ETH3D δ1 ↑ |
|---|---|---|---|---|---|---|---|---|---|---|
| 1 | *DirectAdapt* | FLUX | | | | Direct | 9.7 | 91.2 | 6.0 | 96.0 |
| 2 | *DirectAdapt* | Step1X-Edit | | | | Direct | 9.5 | 91.4 | 5.6 | 96.2 |
| 3 | | Step1X-Edit | ✓ | | | Direct | 8.8 | 93.2 | 5.0 | 97.2 |
| 4 | | Step1X-Edit | ✓ | ✓ | | Direct | 8.6 | 94.0 | 4.8 | 97.3 |
| 5 | | Step1X-Edit | ✓ | ✓ | | Inverse | 6.9 | 95.1 | 4.6 | 98.2 |
| 6 | *Improved* | Step1X-Edit | ✓ | ✓ | | Logarithmic | 6.8 | 95.6 | 3.9 | 98.6 |
| 7 | *Improved* | FLUX | ✓ | ✓ | | Logarithmic | 7.1 | 94.9 | 4.5 | 97.8 |
| 8 | **FE2E** | **Step1X-Edit** | ✓ | ✓ | ✓ | **Logarithmic** | **6.6** | **96.0** | **3.8** | **98.7** |
| 9 | Extension | FLUX-Kontext | ✓ | ✓ | ✓ | Logarithmic | 6.7 | 96.1 | 3.6 | 98.8 |

## 5 CONCLUSION

In this paper, our systematic analysis shows that editors provide a more stable and effective foundation than their generative counterparts. Based on this, we introduced **FE2E**, a novel framework that successfully adapts a pre-trained editing model for dense geometry prediction. To bridge the gap between these tasks, we proposed a consistent velocity training objective for stable convergence and logarithmic quantization to resolve precision conflicts. We also designed a cost-free joint estimation strategy, enabling mutual enhancement within a single forward pass. FE2E achieves SoTA performance and validates the **'From Editor to Estimator'** paradigm, showcasing that harnessing the inherent ability of editing models is an effective and data-efficient approach for dense prediction.

## REPRODUCIBILITY STATEMENT

To ensure the reproducibility of our results, we have included a comprehensive description of the training details, hyperparameters, and training/evaluation datasets in Sec 4.1, 4.2, A.2. The code will be made available upon paper acceptance.

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

## APPENDIX

In this appendix, we provide more implementation details, experiments, analysis, and discussions for a comprehensive evaluation and understanding of FE2E. Detailed contents are listed as follows:

## A  EXPERIMENT SETTINGS

### A.1  AUXILIARY DISPERSION LOSS

Following Diffuse-and-Disperse (Wang & He, 2025), we apply this loss to the output of the 9th block:

$$\mathcal{L}_{\text{disp}} = \log \mathbb{E}_{i,j} \left[ \exp(-\|\eta_i - \eta_j\|_2^2/\tau) \right], \quad (9)$$

where $\eta_{i,j}$ are the output features for the $i$-th and $j$-th samples in a batch, respectively, and

Table 5: Ablation study on Disperse Loss. The baseline is the ID4 model in main paper, Table 4.

| Method | KITTI | | ETH3D | |
|---|---|---|---|---|
| | AbsRel↓ | $\delta1$ ↑ | AbsRel↓ | $\delta1$ ↑ |
| Baseline (CV + FS) | 8.6 | 94.0 | 4.8 | 97.3 |
| + Disperse Loss (DL) | 8.4 | 94.4 | 4.5 | 97.6 |

temperature $\tau = 1$. Finally, the training loss is defined as: $\mathcal{L}_{train} = \mathcal{L}_{fm} + \lambda\mathcal{L}_{\text{disp}}, \lambda = 0.5$. The choices of $\lambda, \tau$, and block all follow the optimal hyperparameters identified in the experiments from Diffuse-and-Disperse.

For integrity, we also conducted ablation studies on this dispersed loss. The performance gains observed in Table 5 confirm that this loss is also effective for dense geometric estimation tasks.

### A.2  EVALUATION DATASETS

We evaluate our model on two tasks: **Zero-shot Affine-Invariant Depth Estimation.** We evaluate on five standard benchmarks: NYUv2 (Silberman et al., 2012), ScanNet (Dai et al., 2017), KITTI (Geiger et al., 2012), ETH3D (Schops et al., 2017), and DIODE (Vasiljevic et al., 2019). Following standard practice, we report the Absolute Relative error (AbsRel) and $\delta_1$ accuracy. **Surface**

**Normal Prediction.** We evaluate on NYUv2, ScanNet, iBims-1 (Koch et al., 2018), and Sintel (Butler et al., 2012) benchmarks. The evaluation metrics are the mean angular error (MeanErr) and the percentage of pixels with an angular error below $11.25°$.

### A.3 EVALUATION METRICS

For zero-shot depth estimation, similar to (Ke et al., 2024), we employ the following evaluation metrics:

- AbsRel: $\frac{1}{|M_{vl}|} \sum_{d \in M_{vl}} |d - d_{gt}| / d_{gt}$;
- $a_1$: percentage of $d$ such that $\max(\frac{d}{d_{gt}}, \frac{d_{gt}}{d}) < 1.25$ ;

where $d_{gt}$ and $d$ denote the GT and estimated pixel depth, $M_{vl}$ is the valid mask (mask rules are consistent with (He et al., 2024)).

For zero-shot normal estimation, we use the following evaluation metrics:

- MeanErr $= \frac{1}{|M_{vl}|} \sum_{\mathbf{n} \in M_{vl}} \frac{180}{\pi} \arccos(\text{clamp}(\mathbf{n} \cdot \mathbf{n}_{gt}, -1, 1))$
- $11.25°$: The percentage of $\mathbf{n}$ where the angular error is less than $11.25°$;

where $\mathbf{n}_{gt}$ and $\mathbf{n}$ denote the GT and estimated normal vector.

## B TRAINING DETAILS OF FINETUNE ANALYSIS

### B.1 IMPROVED EXPERIMENT SETUP

For clarity, we term the direct adaptation of the original editing/generative formulation as *"DirectAdapt"* (Sec 3.2), and Table 4 shows that *DirectAdapt* fails to achieve satisfactory performance. To address this, we introduce two key improvements on training objective (Sec 3.3) and GT quantization(Sec 3.4). They can benefit both editing and generative models, and these *improved* models are better for analyzed our core motivation (Sec 3.1), as they isolate the error from training data and the denoising process. We finally introduce joint training on the editing-based model to obtain **FE2E**.

### B.2 IMPLEMENTATION DETAILS OF GENERATIVE-BASED MODELS

Step1X-Edit is fine-tuned from the generative model FLUX, and both share an almost identical DiT architecture. To further reduce confounding factors, we follow the Step1X-Edit protocol and replace the original FLUX input with a horizontally concatenated noise and RGB image. All hyperparameters, including LoRA settings, optimizer, and training data, are kept exactly the same as those used for FE2E in depth estimation.

FLUX consists of 38 block layers, each producing outputs of consistent dimensions. After rearrangement, the feature map has the shape $B \times 192 \times H/8 \times W/8$, where B is the batch size, H and W are the height and width of the input image. Typically, the output from the final block is projected to 16 channels and passed to the VAE for reconstruction to $B \times 3 \times H \times W$. For visualization, we chose 1, 20, and 35 blocks, operate on the $B \times 192 \times H/8 \times W/8$ feature map, normalize it to $B \times 1 \times H/8 \times W/8$ using the L2 norm, upsample it to $B \times 1 \times H \times W$, and finally visualize it using the Rainbow colormap. The visualization of depth and normals follows the approach of Lotus.

Since our experimental comparisons are conducted using the *improved* model, only one single "denoising" step is performed during inference. Consequently, the output from the VAE decoder directly represents the depth map (the $B \times 3 \times H \times W$ output mentioned before was averaged to obtain a 1-channel depth map), which makes it easier to visualize meaningful features.

## C QUANTIZATION ERROR CALCULATION DETAILS

The following calculations are based on the effective depth range of 0-80m from the Virtual KITTI dataset. The normalization scheme consistently maps an input domain $X$ to the VAE's mandatory

input range of [-1, 1] using the standard min-max scaling formula: $V = 2 \times \frac{X - X_{min}}{X_{max} - X_{min}} - 1$. While other mapping schemes from [0m, 80m] to [-1, 1] may exist, they are not explored in this work. All calculations use the worst-case precision of BF16 over the [-1, 1] interval, which corresponds to a single quantization step of $\Delta V \approx 1/256$.

## C.1 Uniform Quantization

In this scheme, the depth value $D$ is linearly mapped to the [-1, 1] interval. The depth range is $[D_{min}, D_{max}] = [0\text{m}, 80\text{m}]$. The mapping function is $V = 2 \times \frac{D-0}{80-0} - 1 = \frac{D}{40} - 1$. A quantization step of $\Delta V = 1/256$ in the normalized space corresponds to an error $\Delta D$ in the real-world depth space. This error is constant across the entire depth range:

$$\Delta D = 40 \times \Delta V = 40 \times \frac{1}{256} \approx 0.15625$$

At 80m: Error $\approx 16cm$. AbsRel = $\frac{0.16\text{m}}{80\text{m}} = 0.002$.

At 0.1m: Error $\approx 16cm$. AbsRel = $\frac{0.16\text{m}}{0.1\text{m}} = 1.600$.

This method yields an unacceptably large relative error at close distances.

## C.2 Inverse Quantization

This scheme quantizes the reciprocal of depth, i.e., disparity $P = 1/D$. We consider an effective depth range of $[0.1\text{m}, 80\text{m}]$ to avoid division by zero. The corresponding disparity range is $[P_{min}, P_{max}] = [1/80, 1/0.1] = [0.0125, 10]$. The disparity $P$ is linearly mapped to [-1, 1]. The quantization step in disparity, $\Delta P$, is constant:

$$\Delta P = (P_{max} - P_{min}) \times \frac{\Delta V}{2} = (10 - 0.0125) \times \frac{1/256}{2} \approx 0.0195.$$

The relationship between depth error $\Delta D$ and disparity error $\Delta P$ is given by $\Delta D \approx |\frac{d(1/P)}{dP}|\Delta P = \frac{1}{P^2}\Delta P = D^2 \Delta P$.

At 80m: Error = $(80\text{m})^2 \times 0.0195 = 6400 \times 0.0195 \approx 124.8\text{m} \approx 125m$. AbsRel = $\frac{125\text{m}}{80\text{m}} \approx 1.563$.

At 0.1m: Error = $(0.1\text{m})^2 \times 0.0195 = 0.01 \times 0.0195 = 0.000195\text{m} \approx 0.2mm$. AbsRel = $\frac{0.0002\text{m}}{0.1\text{m}} = 0.002$.

As mentioned in the main text, the disparities for 39m and 78m are $1/39 \approx 0.0256$ and $1/78 \approx 0.0128$, respectively. Their difference is $\approx 0.0128$, which is smaller than the disparity quantization step $\Delta P \approx 0.0195$, making them indistinguishable after quantization. This scheme fails completely at large distances.

## C.3 Logarithmic Quantization

This scheme quantizes the logarithmic depth, $D_{log} = \ln(D)$. We again consider the depth range $[0.1\text{m}, 80\text{m}]$. The corresponding log-depth range is $[\ln(0.1), \ln(80)] \approx [-2.30, 4.38]$. The log-depth $D_{log}$ is linearly mapped to [-1, 1]. The quantization step in log-depth, $\Delta D_{log}$, is constant:

$$\Delta D_{log} = (\ln(80) - \ln(0.1)) \times \frac{\Delta V}{2} = (4.38 - (-2.30)) \times \frac{1/256}{2} \approx 0.013.$$

The relationship between depth error $\Delta D$ and log-depth error $\Delta D_{log}$ is given by $\Delta D \approx |\frac{d(e^{D_{log}})}{dD_{log}}|\Delta D_{log} = e^{D_{log}}\Delta D_{log} = D \cdot \Delta D_{log}$. This implies that the absolute relative error, AbsRel $= \Delta D/D$, is approximately constant and equal to $\Delta D_{log} \approx 0.013$.

At 80m: AbsRel $\approx 0.013$. Error = $80\text{m} \times 0.013 = 1.04\text{m}$.

At 0.1m: AbsRel $\approx 0.013$. Error = $0.1\text{m} \times 0.013 = 0.0013\text{m} = 1.3mm$.

This method maintains a reasonable and nearly constant relative error across both near and far ranges, making it a well-balanced and effective solution. The percentile-based normalization used in the main text is a more robust implementation of this fundamental principle.

# D PRELIMINARIES OF FLOW MATCHING

Flow Matching (Lipman et al., 2022) is a highly effective framework for training Continuous Normalizing Flows (CNFs). The core idea is to smoothly transform a simple prior distribution $p_0$ (e.g., the standard Gaussian distribution $\mathcal{N}(0, \mathbf{I})$) into a complex target data distribution $p_1$ over a continuous time variable $t \in [0, 1]$.

This transformation process can be described by an Ordinary Differential Equation (ODE), where the velocity at any time $t$ and point $\mathbf{z}$ is defined by a vector field $v_t(\mathbf{z})$. However, estimating this marginal vector field $v_t(\mathbf{z})$ directly from data samples is challenging. The Flow Matching framework elegantly bypasses this issue by regressing a much simpler and easier-to-compute conditional vector field $u_t(\mathbf{z}|\mathbf{z}_0, \mathbf{z}_1)$ instead.

Specifically, we first sample a pair of points, $(\mathbf{z}_0, \mathbf{z}_1)$, from the prior distribution $p_0$ and the target distribution $p_1$, respectively. We then define a simple path $\mathbf{z}_t$ from $\mathbf{z}_0$ to $\mathbf{z}_1$ and its corresponding conditional vector field $u_t = \frac{d\mathbf{z}_t}{dt}$. It has been proven that if a neural network $f_\theta(\mathbf{z}, t)$ is trained to approximate this simple conditional vector field $u_t$, then in expectation over all sample pairs $(\mathbf{z}_0, \mathbf{z}_1)$ and time $t$, the network $f_\theta$ will converge to the complex marginal vector field $v_t$ that we truly wish to learn.

**Rectified Flow** (Liu et al., 2022) presents a particularly simple and powerful instance of Flow Matching. It defines the path between $\mathbf{z}_0$ and $\mathbf{z}_1$ as a straight line:

$$\mathbf{z}_t = t\mathbf{z}_1 + (1 - t)\mathbf{z}_0, \quad t \in [0, 1].$$

The derivative of this path is trivial, yielding a constant velocity vector that is independent of both time and space:

$$\mathbf{v} = \frac{d\mathbf{z}_t}{dt} = \mathbf{z}_1 - \mathbf{z}_0.$$

Consequently, the training objective (loss function) becomes exceedingly simple: aligning the neural network's prediction with this constant velocity vector $\mathbf{v}$:

$$\mathcal{L} = \mathbb{E}_{t, \mathbf{z}_1, \mathbf{z}_0} \| (\mathbf{z}_1 - \mathbf{z}_0) - f_\theta(\mathbf{z}_t, t) \|^2.$$

**Application in *DirectAdapt*** In this paper, we adapt this framework for a conditional image editing task. Our goal is not to learn an unconditional generative model, but rather a flow from noise $\mathbf{z}_0^y$ to the target geometry latent $\mathbf{z}_1^y$, guided by the input image $\mathbf{x}$ (encoded as $\mathbf{z}^x$). Therefore, our velocity prediction model $f_\theta$ must take $\mathbf{z}^x$ as an additional condition. As shown in Eq. 2 in the main text, our loss function is:

$$\mathcal{L} = \mathbb{E}_{t, \mathbf{z}_1^y, \mathbf{z}_0^y} \| (\mathbf{z}_1^y - \mathbf{z}_0^y) - f_\theta(t, \mathbf{z}^x) \|^2.$$

During inference, we generate the target latent $\hat{\mathbf{z}}_1^y$ by solving the following ODE, with $\mathbf{z}^x$ serving as the guiding condition:

$$\frac{d\hat{\mathbf{z}}_t^y}{dt} = f_\theta(t, \mathbf{z}^x), \quad \text{with initial value } \hat{\mathbf{z}}_0^y \sim \mathcal{N}(0, \mathbf{I}).$$

By integrating from $t = 0$ to $t = 1$ using a numerical ODE solver (e.g., Euler method), we can obtain the final prediction $\hat{\mathbf{z}}_1^y$.

# E REVIEWS OF RELATED GENERATIVE AND EDITING MODELS

The fields of image generation and image editing have always been complementary, and they have undergone several paradigm shifts. The first major breakthrough was the Generative Adversarial Network (GAN) (Goodfellow et al., 2014), which introduced a novel adversarial training process. Then, key advancements in this era include architectural refinements like DCGAN (Radford et al., 2016), the development of conditional and text-to-image GANs such as the StackGAN series (Zhang et al., 2017; 2018), and Cross-Modal Contrastive Learning based models (Zhang et al., 2021). The StyleGAN series (Karras et al., 2019; 2020; 2021) marked a high point for GANs, achieving unprecedented photorealistic high-resolution image synthesis and offering fine-grained control over

visual attributes through a disentangled latent space, which became a cornerstone for many subsequent editing techniques.

More recently, the field has transitioned to Denoising Diffusion Models (Ho et al., 2020), which have become the state-of-the-art for their superior image quality and textual coherence. A series of influential diffusion-based methods were introduced, including GLIDE (Nichol et al., 2021), DALL·E (Ramesh et al., 2021) and its successor DALL·E 2 (Ramesh et al., 2022), Imagen (Saharia et al., 2022), and PIXART-$\alpha$ (Chen et al., 2023). The open-source Stable Diffusion (SD) (Rombach et al., 2022) model, trained on the large-scale LAION-5B dataset (Schuhmann et al., 2022), further democratized high-quality image generation and quickly became a community standard. A growing body of evidence suggests that Diffusion Transformers (Chu et al., 2025; Labs, 2024) outperform U-Nets, motivating the shift toward training modern diffusion models with Transformer architectures.

Building on these powerful generative foundations, the domain of image editing (Lan et al., 2025) (generalized editing) has also advanced rapidly. Early diffusion-based methods like SDEdit (Meng et al., 2022) demonstrated that real images could be edited by adding noise and then denoising with a new text prompt. A significant leap was made with instruction-guided editing, pioneered by InstructPix2Pix (Brooks et al., 2022), which enabled edits based on natural language commands. The field has since diversified with numerous innovative approaches. For instance, DragGAN (Pan et al., 2023) introduced a novel point-based interaction, allowing users to "drag" pixels to precisely deform object shapes. OmniControl (Tan et al., 2025) further enhances controllability by creating a unified framework that accepts diverse spatial guidance signals for both synthesis and editing. This trend towards more powerful and versatile models is also reflected in large-scale systems like UniWorld (Lin et al., 2025), which uses a unified transformer for multi-modal understanding and generation, Step1X-Edit (Liu et al., 2025), fine-tuned from the FLUX architecture for superior instruction following, and multi-modal editors like Qwen-Image (Wu et al., 2025), which leverage Large Language Models (LLMs) to build more comprehensive visual editing frameworks.

## F ADDITION EXPERIMENTS RESULTS

Table 6: Quantitative comparison on zero-shot affine-invariant depth estimation between FE2E and the concurrent unified model.

| Method | NYUv2 (Indoor) | | KITTI (Outdoor) | | ETH3D (Various) | | ScanNet (Indoor) | | DIODE (Various) | | Avg |
| | AbsRel↓ | δ1↑ | AbsRel↓ | δ1↑ | AbsRel↓ | δ1↑ | AbsRel↓ | δ1↑ | AbsRel↓ | δ1↑ | Rank↓ |
|---|---|---|---|---|---|---|---|---|---|---|---|
| Qwen-Image | 5.5 | 96.7 | 7.8 | 95.1 | 6.6 | 96.2 | 4.7 | 97.4 | 19.7 | 83.2 | 2.6 |
| DINOv3 | 4.3 | 98.0 | 7.3 | 96.7 | 5.4 | 97.5 | 4.4 | 98.1 | 25.6 | 82.2 | 1.8 |
| **FE2E** | **4.1** | **97.7** | **6.6** | **96.0** | **3.8** | **98.7** | **4.4** | **97.5** | **22.8** | 81.2 | **1.6** |

**Comparison with Concurrent Unified Model** The field of dense geometry estimation is advancing rapidly, with the task of depth estimation particularly fast. Recently, several concurrent works have been explored to unify the visual tasks, which also include depth estimation benchmarks. As shown in Table 6, our method consistently achieves the top average ranking, even though they are trained with extremely huge data compared to FE2E (e.g., Qwen Image utilizes billions of samples, and DINO v3 is trained on 1.7 billion images).

**Additional Qualitative Comparison** Fig. 9 presents a qualitative comparison between FE2E and other methods. The results demonstrate that our approach produces more refined and accurate depth predictions, particularly in structurally complex regions that may not be fully captured by quantitative metrics. Furthermore, as illustrated in Fig. 10, FE2E consistently delivers precise surface normal predictions, effectively handling intricate geometries and diverse environments. These results highlight the robustness of our method in fine-grained prediction tasks.

## G LIMITATIONS AND FUTURE WORK

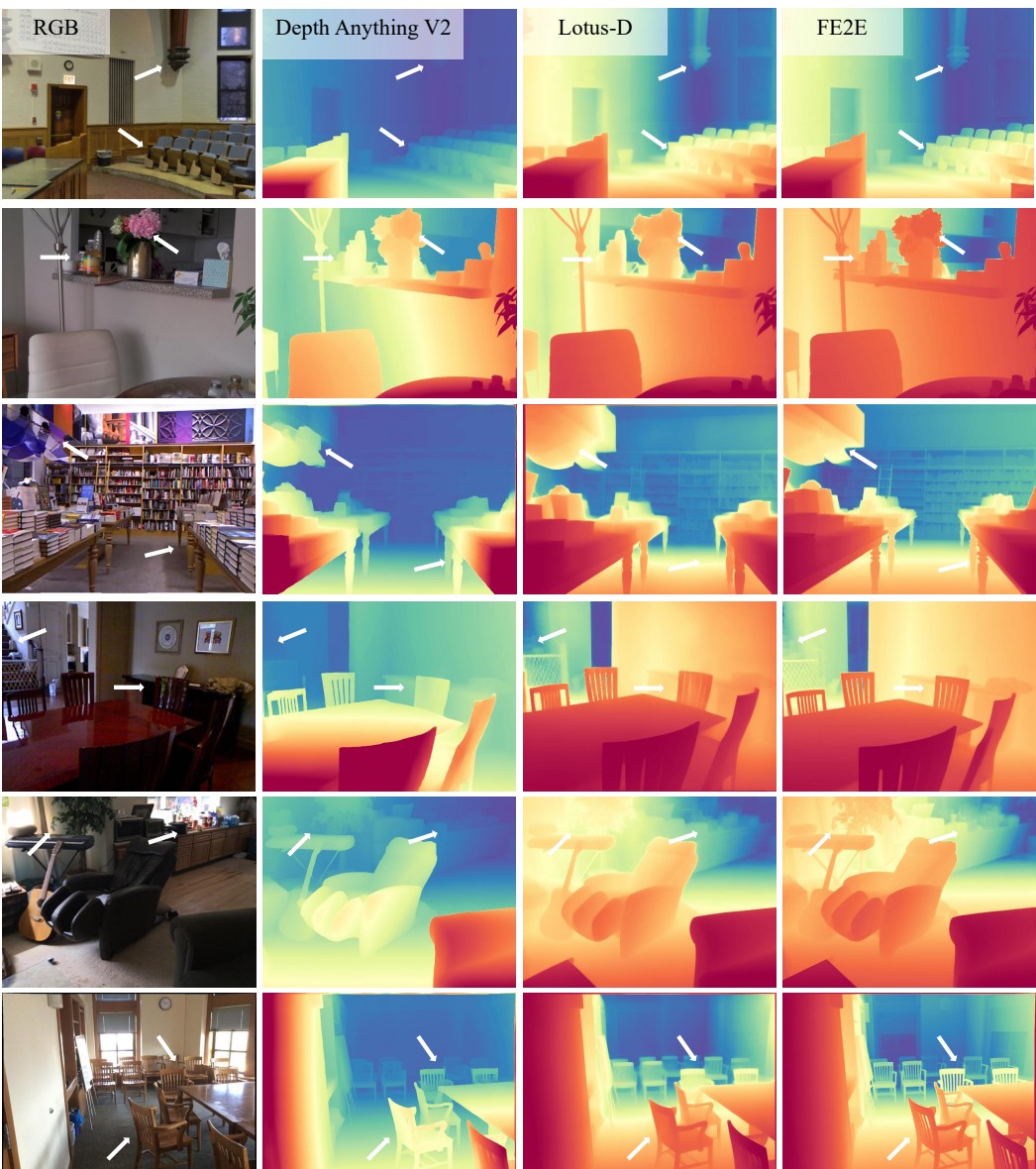

Figure 9: **Additional qualitative comparison on zero-shot affine-invariant depth estimation.** FE2E achieves more accurate depth predictions, particularly in structurally complex regions. White arrows highlight these improvements.

**Large computational load** We present the inference latency and computational complexity of the FE2E model in Table 7, alongside comparisons with previous SD-based and unified methods. Although incorporating DiT does lead to a notable increase in computational complexity relative to other SD-based approaches, FE2E strikes a trade-off between performance and computational efficiency.

Table 7: Performance comparison of different models.

| Methods | Marigold | Lotus-D | Qwen Image | DINO v3 | **FE2E** |
|---------|----------|---------|------------|---------|----------|
| MACs | 133T | 2.65T | 2.13P | 14.5T | **28.9T** |
| RunTime | 9.67s | 212ms | 63.4s | 632ms | **1.78s** |
| AbsRel | 6.5 | 6.1 | 6.6 | 5.4 | **3.8** |

**Diversifying foundation models** The field of image editing is evolving rapidly, and our approach is designed to be model-agnostic. In future work, we plan to incorporate a broader range of editing models to further substantiate the motivation and conclusions presented in this paper.

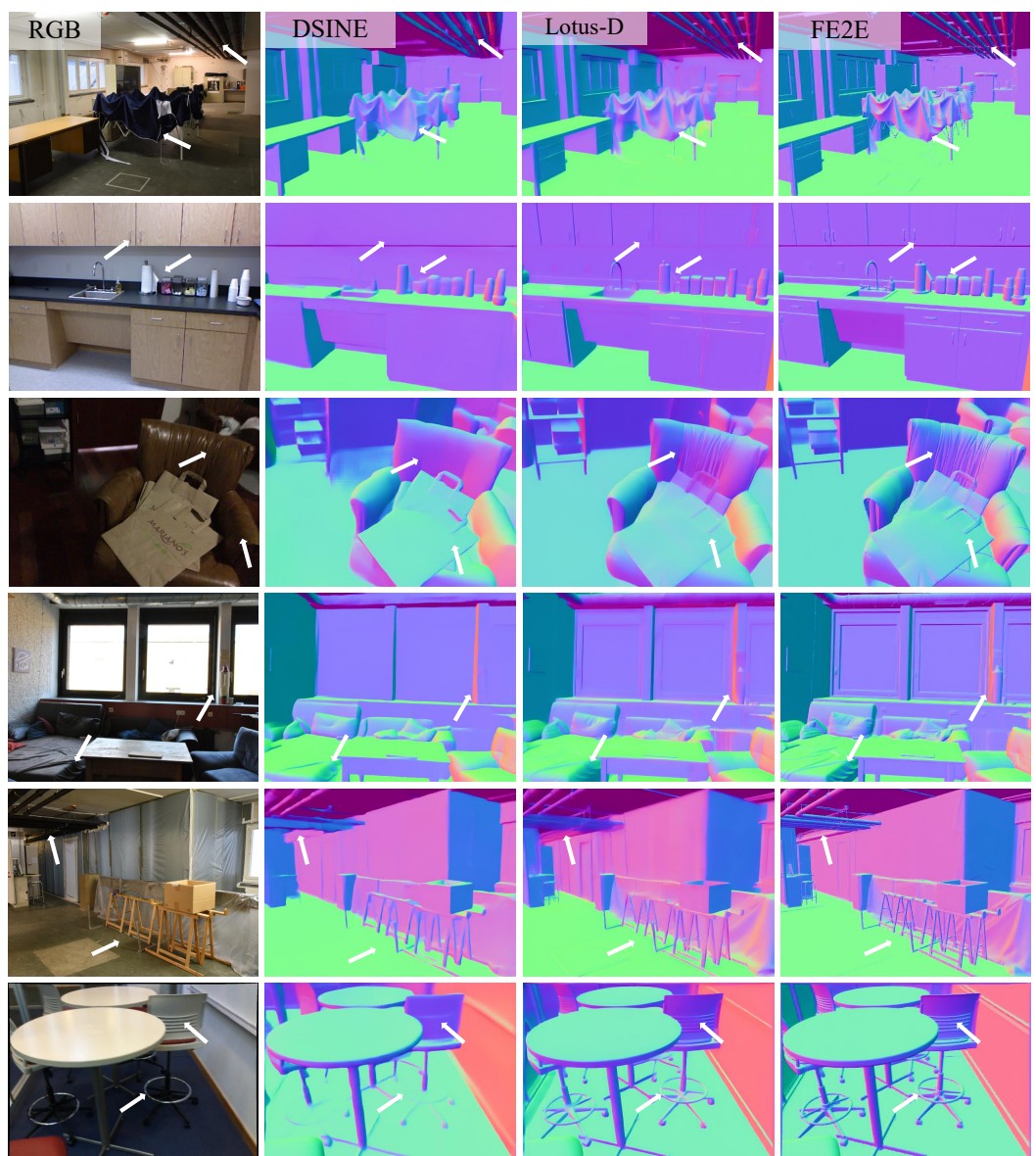

Figure 10: **Additional qualitative comparison on zero-shot surface normal estimation.** FE2E offers improved accuracy, particularly in detailed and complex regions.

**Scaling up the training data**   While a key contribution of this work is demonstrating strong generalization performance with a limited amount of training data, we still anticipate that scaling up the training dataset could further improve the model's capabilities. This direction is meaningful for domains that are not sensitive to computational complexity but require extremely high prediction accuracy. We leave the exploration for future research.

## H   LLM CLARIFICATION

We clarify the use of Large Language Models (LLMs) in the preparation of this manuscript. Specifically, LLMs were employed for language polishing, which involved correcting grammatical errors, improving sentence structure, and enhancing the overall readability and flow of the text.

