# OpenReview forum: "From Editor to Dense Geometry Estimator"
_ICLR.cc/2026/Conference — ICLR 2026 Conference Withdrawn Submission_

### Official Review · Reviewer_zNPE · 2025-10-26

**Soundness:** 1
**Presentation:** 3
**Contribution:** 2
**Rating:** 2
**Confidence:** 5

**Summary:**

This paper introduces FE2E, a novel framework for zero-shot monocular dense geometry prediction by adapting a Diffusion Transformer-based image editing model. The central premise of this work is that image editing models are more suitable for dense geometry prediction tasks than noise-to-image generative models. The authors reformulate the training objective with a Consistent Velocity loss and introduce Logarithmic Quantization to successfully convert the editor into a deterministic estimator for depth and surface normals. FE2E achieves state-of-the-art performance on various zero-shot benchmarks.

**Strengths:**

1. The paper innovatively adapts the pre-trained image editing model for dense geometry prediction, rather than the noise-to-image generative models like other diffusion-based methods.

2. The paper achieves superior quantitative results across multiple benchmarks.

**Weaknesses:**

Although the final performance is promising, the paper still lacks solid justifications for the key designs. The statement "editing models possess inherent structural priors" is empirical and insufficient to support acceptance.

1. The core claim that "editing models are fundamentally better for dense geometry prediction" is intuitive, but the evidence provided is weak.
- Step1X-Edit is fine-tuned from FLUX, is the performance gain due to Step1X-Edit's additional fine-tuning and data resource?
- In Fig.3, what is the specific value of the time-step $t$ used for the FLUX-based method? If a very small $t$ (e.g., $t = 0$) is used, the DiT's input in the FLUX model would be close to the clean image. This makes the FLUX-based method also operate like an image-to-image translation in your setting. In this case, does the FLUX-based method still exhibit "abstract and unstructured" features?

2. Why the time-step t can be remove in Eq. 4, is there any theoretical evidance? Furthermore, given that the base Step1X-Edit model relies on $t$, how do you deal with it during finetuning? Also, why does $z_{0}^{y}=0$ enable greater consistency?

3. Does Fig. 5 consider the input image $z^x$ (the condition of the editing model)? If so, why is there only a single data point shown on the left of Fig. 5(c)?

4. Sec. 3.4 requires a clearer explanation to justify the logarithmic method. Why the $\pm[0.5, 1.0]$ interval is the worst-case for BF16 and why the RGB (also be normalized to [-1,1]) succeeds while the depth map fails? The analysis of the proposed logarithmic method needs more detailed theoretical explanation for why it is inherently superior to both the uniform and inverse methods.

5. The cost-free joint estimation is not novel. For example, GeoWizard also connects the depth and normal estimation via the cross-domain self-attention, and DICEPTION also utilizes the DiT advantage for multi-tasks.

6. Aside from utilizing the pre-trained editing model, how does your method differ from other single-step generation-based approaches?

**Questions:**

See the weaknesses.
If the authors can address my concerns, I am willing to change my score.

---

> ### Author Response · Authors · 2025-11-13
>
> We respectfully appreciate your time and feedback on our manuscript. However, we would like to argue that **your comments and score are unfair** to our paper, as many of the issues raised are already clearly addressed in the manuscript. However, we are glad to see your willingness to reconsider your score.
>
> `W1`, you stated that our claim, "editing models possess inherent structural priors," is merely empirical. If you refer to `Section 3.1 and Figure 3`, we provide a **detailed visual analysis** demonstrating the structural features at the input-end of the editing model, which are absent in standard generative models. The result in Table 4 also give the experimental results. We believe these supports a conclusion that goes beyond empiricism alone.
>
> - regarding the advantage being due to more data: The training data for FLUX is unpublished, but its performance surpasses SD3, suggesting its training data is at least on the same order of magnitude (i.e., billion-scale[1]). In contrast, the Step1X-Edit paper states it was trained on a million-scale data[2]. Is it not plausible that a model, after receiving only an additional **0.1%** of training, suddenly gains a superior fine-tuning characteristic. We believe the advantage of Step1X-Edit over FLUX lies in the fact that the I2I paradigm of the edited model is more suitable for dense geometry pridiction and has the 3 advantages during fine-tuning as analyzed in `Section 3.1`.
>
> - Please note that both FLUX and Step1X-Edit use the strategy of starting from **zero noise**, with **no add-noising step**; therefore, the $t$ setting is completely irrelevant. Furthermore, both methods use the exact same concatenation input, i.e., the input is always $[0, z^x]$. Both models can see the clear $z^x$, yet FLUX still exhibits abstract and unstructured features (these details are already provided at `L192 and Sec B.2`).
>
> `W2`  As mentioned above, the $t$ setting for Step1xEdit is irrelevant theoretically, as we stress that our method is **independent of $t$**. Our experiments also show that different $t$ values have minimal impact on the final result. For elegance and to emphasize this independence, we described it as 'not using $t$'.  The $t$ value for our reported model was set to 1.
>
> `W3` Regarding Figure 5, the single data point at the start ($t=0$) is meant to represent **determinism**, in contrast to the **uncertainty** of starting from random Gaussian noise. The input image itself is just a deterministic input, meaning that for any given input, there is only one unique GT. We acknowledge this figure may have caused a misunderstanding.
>
> `W4` Regarding Section 3.4, we invite you to check the **BF16 formulation`L292`**:
> - the precision between [0.5, 1.0] is $2^{126-127} \times 2^{-7}=1/{256}$ (same with `L293-298`)
> - Between [0.25, 0.5] is $2^{126-127} \times 2^{-8}=1/{512}$
> - Between [0.125, 0.25] is $2^{126-127} \times 2^{-9}=1/{1024}$, and so on.
>
> The reason RGB succeeds is clearly stated in the paper `L299`: even the worst-case precision (1/256) is sufficient for the 0-255 image pixel value range.
>
> - We believe our error calculations at different depth values mathematically demonstrate the **disadvantages** of the other two methods. The huge error of linear quantization at low depths and the huge error of inverse quantization at high depths clearly illustrate the superiority of our logarithmic method (`Table 1 and Figure 6`). The calculation are already details in `L315, Sec C`.
>
> `W5` Regarding joint estimation: We have explicitly stated in the paper (`L347`)that GeoWizard, which you mentioned, requires **extra computational overhead and a switcher**, and our method is cost-free. As for Diception, our motivation is fundamentally different; we use it to achieve mutual promotion and synchronous inference between tasks.
>
> `W6` Regarding your point on novelty :
> We must state that introducing the editing model is the core concept of this paper. We have provided extensive analysis and comparative experiments to support this; excluding it from the evaluation of novelty is unreasonable. Even if this core contribution were excluded, our method still offers significant novelty. Compared to other models, our method  consistency velocity can be seen as an extension of the single-step denoising paradigm (traditionally relies on noise loss) to the flow-matching framework. This adaptation of flow-matching has not been discussed in prior work.
> Furthermore, the quantization precision is also critically important. As stated in the paper, an increasing number of models are being trained using BF16, and this is a significant future trend. This creates a direct conflict with the inherent high-precision demands of depth estimation. Our logarithmic quantization method effectively alleviates this conflict.
> Therefore, even if you do not consider the joint estimation to be innovative, we believe this paper still possesses at least three other sufficient contributions.

---

> ### Author Response · Authors · 2025-11-13
>
> We hope for your prompt response.
>
> [1]Stable Diffusion 3: Research Paper
>
> [2]Step1X-Edit: A Practical Framework for General Image Editing

---

### Official Review · Reviewer_ZQbZ · 2025-10-27

**Soundness:** 4
**Presentation:** 3
**Contribution:** 3
**Rating:** 6
**Confidence:** 4

**Summary:**

This paper introduces FE2E (From Editor to Dense Geometry Estimator), a novel framework for monocular depth and normal estimation. The core hypothesis is that pretrained image-to-image (I2I) *editing* models serve as a more suitable foundation for this I2I task than common text-to-image (T2I) *generative* models. The authors support this by comparing an editor to its generative base, showing the editor possesses more aligned geometric features. To adapt the DiT-based editor, the paper proposes three key technical contributions: (1) a "Consistent Velocity" flow matching objective with a fixed origin, which simplifies the deterministic prediction task; (2) a "Logarithmic Annotation Quantization" scheme to resolve the precision conflict between the model's native BFloat16 format and the high-precision demands of geometry estimation; and (3) a "Cost-Free Joint Estimation" method that leverages the DiT's architecture to predict depth and normals simultaneously without extra computation. Experiments show that FE2E achieves state-of-the-art zero-shot performance.

**Strengths:**

The paper's primary strength lies in its novel and well-motivated premise of using image editing priors for dense estimation. This I2I-for-I2I paradigm shift is a novel conceptual contribution. This claim is well-supported by a systematic comparative analysis in Section 3.1, which convincingly illustrates that the editor model's features are inherently better structured for this task and lead to a more stable finetuning process. The technical contributions designed to bridge the gap between editing and estimation are solid and effective. The analysis of the BF16 precision conflict (Sec 3.4) is particularly thorough, and the proposed logarithmic quantization is a clever and necessary solution . The reformulation of the flow matching loss for a deterministic task (Sec 3.3) and the computationally elegant joint-estimation strategy (Sec 3.5) are also strong contributions. These elements combine to produce impressive state-of-the-art zero-shot results across multiple benchmarks, validating the data-efficient "From Editor to Estimator" approach.

**Weaknesses:**

W1: Line 84. "However, these generative models are initially designed for T2I generation and lack the ability to capture the geometric cues from the absent image inputs." This motivation seems overstated. Recent T2I-based approaches (e.g., Marigold) have demonstrated strong robustness in depth estimation, suggesting that T2I priors can, in practice, encode meaningful geometric cues. We recommend softening the claim and positioning the contribution as complementary.

W2: In sec 3.1, we suggest the authors to briefly introduce the background and overall task setting of the experiments before describing the experimental setup details. Providing this high-level context would help readers better understand the paper.

W3：The "Auxiliary Dispersion Loss" ($\mathcal{L}_{disp}$) is only introduced in Appendix A.1. Table 5 demonstrates its non-trivial performance contribution, and it is part of the final training objective. Omitting this from the main methodology section (Sec 3) makes the description of the final, best-performing model (ID8) incomplete and potentially misattributes the source of its full performance.

W4：The analysis of the "cost-free joint estimation" mechanism (Sec 3.5) is somewhat shallow. The paper states that the DiT's global attention "naturally allows" for information exchange and that this enables "mutual enhancement", but it does not provide a deeper analysis (e.g., attention visualization) to show *how* or *where* this information is exchanged and leveraged. The empirical gain from this component (ID6 vs. ID8 in Table 4) appears positive but modest, making it hard to assess the true impact of this synergy.

**Questions:**

Q1: How dependent are the proposed contributions, particularly the "Consistent Velocity" objective (which simplifies inference to a single step ) and the "Cost-Free Joint Estimation" (which relies on specific DiT input concatenation), on the DiT architecture? Could these ideas be generalized to other editor architectures, such as those based on U-Nets?

Q2: I am a bit confused by the "consistent velocity" reformulation in Section 3.3. The original flow matching (Eq. 2) predicts velocity $f_{\theta}(z^{x},z_{t}^{y},t)$, which is time-dependent. The new formulation (Eq. 4) removes the time $t$ dependency and (Eq. 5) removes the $z_{0}^{y}$ dependency , ultimately simplifying inference to $z_{1}^{y} = f_{\theta}(z^{x})$. This seems to imply the model is no longer a flow-matching model but a simple regressor. Could the authors clarify if this interpretation is correct and why this is still framed as "flow matching

---

### Official Review · Reviewer_ZMF9 · 2025-10-29

**Soundness:** 3
**Presentation:** 3
**Contribution:** 3
**Rating:** 4
**Confidence:** 4

**Summary:**

This submission introduces FE2E, a novel framework that pioneers adapting DiT-based image editing models (e.g., Step1X-Edit) for zero-shot dense geometry prediction (depth/normal estimation), addressing limitations of existing T2I generative model-based approaches.

Motivated by the misalignment of T2I models with I2I dense prediction, the authors systematically show editing models possess inherent structural priors—their fine-tuning "refines" rather than reshapes features, enabling more stable convergence and higher performance than generative counterparts (e.g., FLUX). To adapt editors for deterministic geometry tasks, FE2E reformulates flow matching loss into a "consistent velocity" objective (fixed) to eliminate trajectory errors, uses logarithmic quantization to resolve BF16 precision conflicts, and leverages DiT’s global attention for cost-free joint depth/normal estimation.

Trained on only 71K synthetic samples (Hypersim + Virtual KITTI), FE2E achieves SOTA results: 35% AbsRel reduction on ETH3D, 10% on KITTI, and outperforms data-hungry models like DepthAnything (62.6M data) across 5 depth/4 normal benchmarks. Ablations confirm each component’s value—e.g., consistent velocity cuts ETH3D error by 10%, and joint estimation enhances cross-task supervision.

**Strengths:**

1. The work conducts rigorous, comparative empirical experiments to substantiate that image editing models are more suitable as priors for dense geometry prediction than traditional image generation models—an intuition-aligned conclusion. By selecting Step1X-Edit (editor) and FLUX (generator, sharing DiT architecture for fairness) as baselines, the authors analyze feature evolution, training dynamics, and final performance: early-stage editor features already align with input geometric structures (vs. unstructured generative features), editor fine-tuning “refines” rather than reshapes features (avoiding generative models’ unstable convergence and loss bottlenecks), and editing-based models consistently outperform generative counterparts (e.g., Step1X-Edit-based models reduce ETH3D AbsRel to 3.8 vs. FLUX-based 4.5). These experiments not only validate the intuitive advantage of editing models for I2I dense tasks but also provide clear, reproducible evidence to support the paradigm shift from generators to editor.

2. To resolve the precision conflict between image editors’ native BF16 format (sufficient for RGB but inadequate for high-precision depth/normal estimation), the authors propose a logarithmic quantization scheme—an innovative solution tailored to geometric prediction needs. Unlike uniform quantization (causing ~0.16m constant error, 1.6 AbsRel at 0.1m) or inverse quantization (failing at long distances, e.g., 39m/78m indistinguishable), logarithmic quantization applies and percentile-based normalization, maintaining balanced, low relative error (AbsRel≈0.013) across near (0.1m) and far (80m) ranges. This design not only preserves BF16’s computational efficiency (avoiding FP32’s cost) but also eliminates precision-related artifacts, directly contributing to FE2E’s SOTA performance (e.g., 19% KITTI AbsRel reduction vs. uniform quantization). This is useful for the other research that need to use FP16 models as prior model.

**Weaknesses:**

The paper’s formulation of the "consistent velocity" training objective, while framed as an innovation, lacks novelty and ultimately reduces to an operation already mentioned in prior work (e.g., Lotus). The authors reformulate the editor’s original flow matching loss by fixing the starting point t_0=0 and enforcing velocity independence from time t, leading to the inference simplification:  (Eq. in Sec 3.3). This derivation effectively equates the final geometric latent to the DiT’s direct output conditioned on the input image’s latent
—a result functionally equivalent to directly predicting the target (e.g., depth/normal) from the input, rather than learning a "flow" from a noise origin. Critically, this "fixed starting point + direct output" logic was already explored in Lotus (He et al., 2024), where the model leverages conditional inputs to predict geometric targets without relying on stochastic noise trajectories. The paper fails to acknowledge this overlap or demonstrate how its "consistent velocity" differs from such prior deterministic prediction paradigms, weakening the novelty of this component.


Also, the "consistent velocity" framework suffers from inconsistent input handling for joint depth and normal estimation, as visualized in Fig. 2 and contradicted by the paper’s own formulas. Per the formulation, the flow should start from a fixed origin for all geometric targets (depth and normal). However, Fig. 2 shows depth is derived from RGB image tokens while normal is derived from the fixed zero input—creating a discrepancy that violates the "consistent" premise of the velocity objective. This inconsistency raises critical questions: if the framework claims to unify geometric prediction via a fixed starting point, why are depth and normal treated differently in input sourcing?
Further, the paper provides no experimental validation to clarify how input choices impact performance—an oversight that undermines the credibility of the "consistent velocity". For instance:

There is no test of swapping inputs (e.g., depth from zero, normal from RGB tokens) to verify if the current design is optimal, or if the input-source difference drives performance gaps between depth and normal results.

The paper does not explore uniform input conditioning (e.g., using RGB for both depth and normal, with positional encoding (PE) to distinguish tasks) to test if consistency in inputs improves joint estimation synergy.

There is no ablation on learnable input tokens (e.g., analogous to LRM (Liu et al., 2023)) to compare fixed zero inputs against adaptive tokens, which could reveal whether the fixed origin is truly superior or merely a suboptimal simplification.

Without these experiments, the paper cannot substantiate that its "consistent velocity" design is robust, or that its input discrepancy does not bias results—leaving the core objective’s validity.

So base on above, I think authors should conduct experiments in the rebuttal: 1.swapping inputs  2. uniform input conditioning  3.learnable input tokens as input.

**Questions:**

Same as Weaknesses. Since I suspect that if  the CONSISTENT VELOCITY  design is robust, I give this paper with 4. The whole paper is good and provide several useful conclusions, so if the author can solve the problem I posted, I will rise my score and lead to accept this paper.

---

### Official Review · Reviewer_7kH3 · 2025-10-31

**Soundness:** 3
**Presentation:** 3
**Contribution:** 3
**Rating:** 4
**Confidence:** 4

**Summary:**

This paper argues that image editing models are better than generative models for geometry estimation tasks like depth. It introduces FE2E, a framework that adapts an editor model for this task using three key techniques: a new 'Consistent Velocity' loss, a 'Logarithmic Quantization' method to fix precision issues, and a 'Cost-Free Joint Estimation' strategy. Despite using 100x less data (71K vs 62.6M), FE2E outperforms models like DepthAnything in zero-shot benchmarks.

**Strengths:**

1. The core argument is novel and well-supported: The central idea of the paper—that image editors (which must understand input structure) are better suited for I2I prediction tasks than T2I generators—is intuitive and insightful. The feature visualizations and the ablation studies provide strong evidence for this hypothesis.
2. Technical contributions are targeted and effective: This paper doesn't just fine-tune a model; it astutely identifies the key mismatches between the editing model and the estimation task (deterministic vs. stochastic; BF16 vs. high-precision). It proposes "Consistent Velocity" and "Logarithmic Quantization" as two specific and efficient solutions. The analysis of logarithmic quantization is particularly excellent.
3. Experimental results are impressive: The model achieves SOTA performance on multiple benchmarks (notably a 35% AbsRel improvement on ETH3D). Surpassing data-driven models like DepthAnything with only 0.2% of the data is a powerful demonstration of the data efficiency gained from leveraging the correct inductive bias.

**Weaknesses:**

1. Prohibitively High Computational Cost: This is the method's most significant drawback. The model requires 28.9T MACs and 1.78s for inference. This is an order of magnitude higher than diffusion-based competitors like Lotus-D (2.65T MACs, 212ms). While the authors acknowledge this trade-off, this latency and cost make the model impractical for most real-world applications (e.g., robotics, AR/VR) and situate it purely as an academic "best-performer" at any cost.
2. Limited Generalizability of the Core Premise: The paper's foundational claim—that "editors are better than generators"—is supported only by a direct comparison between Step1X-Edit and FLUX. While these share a DiT architecture, this represents a single data point (n=1 pair). This conclusion is an overclaim, as it's unclear if this finding would generalize to other model pairs, particularly the widely-used U-Net architectures (e.g., InstructPix2Pix vs. Stable Diffusion). The paper's claim is only truly supported for this specific DiT-based model family.
3. Unexplored Data Scaling Properties: The paper's primary narrative is built on data efficiency, showing SOTA results with only 71K images. However, the paper explicitly avoids exploring how FE2E itself performs when scaled to larger datasets. This is a critical omission. Does the model's performance saturate quickly (making it truly "data-efficient"), or would it also benefit massively from the 62.6M images used by competitors? This unanswered question makes it difficult to fully assess the model's true data-efficiency profile.
4. Conflated Ablation Study: The ablation study does not cleanly disentangle the impact of the three main contributions. For instance, the "Logarithmic Quantization" appears to provide the single largest performance boost. However, its effect is only measured after "Consistent Velocity" (CV) and "Fixed Start" (FS) are already applied. There is no baseline showing the effect of only DirectAdapt + LogQuant. This makes it difficult to ascertain the true individual importance of each component, and it suggests that the quantization strategy might be the dominant factor in the performance gain, more so than the novel flow matching loss.
5. Weak Rationale and Marginal Gain for Joint Estimation: The "Cost-Free Joint Estimation" is presented as a key contribution, but its justification is weak and its impact is minimal. The paper does not provide a strong intuition for why supervising the output p_l (which corresponds to the conditional latent $z^x$) should work. Furthermore, the empirical gain from this component is marginal; adding Joint Estimation only improves the ETH3D AbsRel from 3.9 to 3.8. This suggests it is a minor refinement rather than a significant contribution.

**Questions:**

1.Given the model's massive computational overhead, have the authors explored any model compression techniques, such as distillation or pruning? Is it possible to distill FE2E into a more lightweight U-Net architecture (similar to Lotus-D) to achieve practical inference speeds while retaining most of the accuracy benefits?
2. Can the authors comment on why data scaling experiments were not performed? What is your hypothesis on how FE2E would perform if it were also trained on the 62.6M images used by DepthAnything?
3. To better understand the contributions, could you provide results for an ablation that applies Logarithmic Quantization directly to the DirectAdapt baseline? This would help clarify if the quantization strategy or the consistent velocity objective provides the majority of the performance gain.
4. Could the authors elaborate on the intuition behind the "Cost-Free Joint Estimation"? Why does supervising the output side p_l, which corresponds to the conditioning latent $z^x$, help the model learn? Did you test other, more traditional joint estimation methods (e.g., adding two separate prediction heads after the DiT) as a baseline?

---

> ### Author Response · Authors · 2025-11-14
>
> We respectfully appreciate your time and feedback on our manuscript. Here is my response:
>
> `W1` We acknowledge that the FLUX-based methods introduce higher computational overhead and inference latency. However, we wish to contextualize this issue within our contribution and the current research trends.
>
> - The primary objective of this paper is not to deliver a lightweight, production-ready model. Rather, our goal is to be the **first** to systematically investigate and validate that **Editors are superior to Generators** as foundation models for dense geometry estimation. Consequently, the high computational cost is an **inherent property** of the SoTA DiT backbone we chose, not an overhead introduced by our FE2E adaptation framework. Our contribution lies in demonstrating this new paradigm's vast potential in performance and, critically, outperforming DepthAnything with only 0.2% of its data.
>
> - Recently, a series of leading works represented by **Marigold (CVPR 2024 Best Paper Candidate), Lotus(ICLR 2025) and DepthAnything (CVPR/NIPS 2024)** have established a research paradigm within the community: **there is a willingness to accept a certain level of computational cost at this stage in exchange for the unprecedented generalization and quality afforded by Large Vision Models(LVMs).**  Marigold, for example, is also 10x more expensive than traditional methods like MiDaS, yet its exceptional zero-shot capability led to its widespread acceptance. Our work shares this exact philosophy: we explore a powerful new source of LVM priors.
>
> `Q1` Regarding the model compression, we must firmly state that this is **irrelevant to our paper's core contribution.**
>
> Our paper focuses on answering a fundamental research question about *editing model for geometry estimation*. Model compression is entirely an irrelevant topic. While valuable, it has no bearing on the validity of our conclusion that the editor's priors are more suitable for this task.
>
> `W2` we must **respectfully disagree with the characterization of our evidence as a "single data point (n=1)."** We kindly ask the reviewer to please re-examine `Sec 4.4 and Table 4.`
>
> - **Besides Step1X-Edit, we also validate other DiT Editors**
> - Our primary controlled experiment compares **FLUX (Generator, ID7)** against **Step1X-Edit (Editor, ID6)**, as they share the same DiT architecture. This comparison (detailed in Sec 3.1, Fig 3, Fig 4) establishes the superiority of the editor's priors.
>    - **Crucially**, to *explicitly* test the generalization of our findings, we included **ID9 (Extension)**. In this experiment, we applied our identical adaptation protocol to **FLUX-Kontext**, a different and concurrent SoTA DiT-based editor.
>    - The results from FLUX-Kontext (ID9) were fully consistent with our main FE2E model (ID8) .
>    - Theoretically, our methods can generlize to any DiT-based editor, as they share the same DiT base architecture.
>
> `W3`, we believe the implied request for a 62.6M-image scaling experiment is **unreasonable** and different from the definition of data efficiency in this paper.
>
> -  Massive datasets usage is contrary to our motivation: The central motivation of our work is to demonstrate that **by harnessing the powerful priors of editing models, we can achieve SoTA performance using only 71K images**— 0.2% of the data used by competitors like DepthAnything. Conducting the 62.6M-image experiment would **obscure this contribution** and shift the narrative of "prior knowledge" investigation.
>
> - The statement of "data efficiency" have never appear in our paper.  But in this paper, it stands for **our method can achieve a comparable or superior result to a SoTA competitor while using over limited data.**
>
> `W4` we can commit to adding a more comprehensive ablation study to the future version of our manuscript.
>
> `W5` We will **restate this contribution in our final manuscript** to address this. The primary motivation of our "Cost-Free Joint Estimation" (Sec 3.5) is **not performance boost, but efficiency.**
>
> `Q4` (The Intuition): Our method is based on a simple insight:
>
> 1. The DiT editor processes a [z^x, z^y] input and produces a [p_l, p_r] output.
> 2. In the original design, the entire p_l output region is **discarded**.
> 3. We **re-purpose this discarded p_l region** to supervise a second task.
>
> The true benefit of this component is that it allows us to **predict two tasks (e.g., depth and normal) in a single forward pass, without any additional cost. **The minor but consistent performance gain is simply an extra"bonus".
>
> We hope for your prompt response.

---

### Note · Authors · 2025-11-14

I have read and agree with the venue's withdrawal policy on behalf of myself and my co-authors.